# Cytokines from Bench to Bedside: A Retrospective Study Identifies a Definite Panel of Biomarkers to Early Assess the Risk of Negative Outcome in COVID-19 Patients

**DOI:** 10.3390/ijms23094830

**Published:** 2022-04-27

**Authors:** Martina Fabris, Fabio Del Ben, Emanuela Sozio, Antonio Paolo Beltrami, Adriana Cifù, Giacomo Bertolino, Federica Caponnetto, Marco Cotrufo, Carlo Tascini, Francesco Curcio

**Affiliations:** 1Laboratory of Immunopathology, Institute of Clinical Pathology, Department of Laboratory Medicine, University Hospital of Udine, 33100 Udine, Italy; antonio.beltrami@uniud.it (A.P.B.); francesco.curcio@uniud.it (F.C.); 2Dipartimento di Area Medica (DAME), University of Udine, 33100 Udine, Italy; delben.fabio@spes.uniud.it (F.D.B.); adriana.cifu@uniud.it (A.C.); federica.caponnetto@uniud.it (F.C.); 3Clinic of Infectious Diseases, University Hospital of Udine, 33100 Udine, Italy; emanuela.sozio@gmail.com (E.S.); marcocotrufo@gmail.com (M.C.); carlo.tascini@asufc.sanita.fvg.it (C.T.); 4Pharmacy Department, Azienda Ospedaliero-Universitaria di Cagliari, 09042 Cagliari, Italy; giacomo.bertolino1985@gmail.com

**Keywords:** COVID-19, cytokine, proadrenomedullin, IL-6, IL-10, sIL2Rα, IP10, prognosis

## Abstract

The main aim of this study was to identify the most relevant cytokines which, when assessed in the earliest stages from hospital admission, may help to select COVID-19 patients with worse prognosis. A retrospective observational study was conducted in 415 COVID-19 patients (272 males; mean age 68 ± 14 years) hospitalized between May 2020 and March 2021. Within the first 72 h from hospital admission, patients were tested for a large panel of biomarkers, including C-reactive protein (CRP), Mid-regional proadrenomedullin (MR-proADM), Interferon-γ, interleukin 6 (IL-6), IL-1β, IL-8, IL-10, soluble IL2-receptor-α (sIL2Rα), IP10 and TNFα. Extensive statistical analyses were performed (correlations, *t*-tests, ranking tests and tree modeling). The mortality rate was 65/415 (15.7%) and a negative outcome (death and/or orotracheal intubation) affected 98/415 (23.6%) of cases. Univariate tests showed the majority of biomarkers increased in severe patients, but ranking tests helped to select the best variables to put on decisional tree modeling which identified IL-6 as the first dichotomic marker with a cut-off of 114 pg/mL. Then, a good synergy was found between IL-10, MR-proADM, sIL2Rα, IP10 and CRP in increasing the predictive value in classifying patients at risk or not for a negative outcome. In conclusion, beside IL-6, a panel of other cytokines representing the degree of immunoparalysis and the anti-inflammatory response (IP10, sIL2Rα and IL-10) showed synergic role when combined to biomarkers of systemic inflammation and endothelial dysfunction (CRP, MR-proADM) and may also better explain disease pathogenesis and suggests targeted intervention.

## 1. Introduction

Severe Acute Respiratory Syndrome Coronavirus 2 (SARS-CoV-2) is the etiologic agent of a syndrome (COVID-19) that evolves from a viral replication phase to an inflammatory phase, that may be further complicated with secondary infections, evolution to a multisystem inflammatory phase or to a post-acute phase characterized by immune-paralysis [1]. The initial inflammatory phase, that has been compared to a cytokine storm (i.e., a severe inflammatory syndrome characterized by immune cell hyperactivation and increased levels of circulating cytokines triggered by different pathologic conditions, including infections), negatively impacts the outcome of COVID-19 patients [2]. Although not all the authors agree with the cytokine storm, undoubtedly elevated proinflammatory cytokines are hallmarks of severe SARS-CoV2 infection. Notably, the immune response to the pathogen, and not the pathogen itself, seems to be responsible for the exaggerated release of inflammatory molecules during COVID-19 infection. However, one should consider that the distinction between the normal response to a severe infection and an excessive response is not always clear, since the same cytokines that could be helpful in limiting the infectious agent may be harmful for the patient as well [2]. Interleukin (IL)-1, IL-6, IL-8, tumor necrosis factor-α (TNFα) and interferon-gamma (INFγ) play a pivotal role in the pathogenesis of the cytokine storm syndrome. In COVID-19 patients, IL-6 and TNFα are among the most important ones, given that their levels are strong independent predictors of patient survival [3]. However, other cytokines were frequently described in the literature as increased during the COVID-19 inflammatory burst and variably identify patients with worse prognosis and/or in a different phase of the pathological process. Consistently, interferons (the classic markers of anti-viral response with their related chemokines CXCL9 and CXCL10/IP10), may be evaded by SARS-CoV2, being dysregulated or inefficient in a subset of patients [2,4,5]. In fact, lower basal circulating interferon plasma levels appeared as a risk factor for delayed viral clearance, severe COVID-19 and lung fibrosis [6,7]. SARS-CoV2 causes direct injury in the lung early on during its natural history, triggering the NLRP3 inflammasome that leads to both endothelial and epithelial damage and starts the extensive inflammatory and fibrotic processes [5]. Initially, after the lung injury, resident macrophages are polarized to the M1 phenotype by IFNγ and toll-like receptor (TLR) ligands and contribute to the host defense by generating reactive oxygen species (ROS) and reactive nitric oxide (NO) and by releasing pro-inflammatory cytokines and chemokines (e.g., IL-1β, IL-12, IL-23, TNFα and CCL2) [8,9]. IL-1β and TNFα are also potent activators of neutrophils that are attracted by IL-8 and release large amount of neutrophil extracellular traps (NETs) that are present in the sera of COVID-19 patients and are correlated with both progressive loss of lung functionality and thrombosis [10,11]. Severe COVID-19 is characterized by the accumulation of profibrotic monocyte-derived macrophages, whose phenotypic reprogramming may be induced by the virus itself, according to recent data [12]. In this context, TGFβ is a crucial factor that sustains all the phases of the fibrotic process, promoting the differentiation of myofibroblast from progenitors and the generation of excess extra-cellular matrix components [12]. Another important actor that deserves attention is IL-10, since elevated levels predict poor outcomes in patients with COVID-19 [13]. Lastly, a prominent elevation of soluble IL-2 receptor alpha (sIL2Rα, also called as soluble CD25), a usual marker of T-cell activation, characterizes patients in whom T cell exhaustion favors lung destruction in the late phase of disease evolution [14,15,16]. In addition, inflammation, independent groups, including ours, have recently focused their research on vascular damage. In this context, mid-regional pro-adrenomedullin (MR-proADM) is an emerging marker of sepsis that plays a critical role in the regulation of vascular permeability and stabilization of microcirculation, in the response to tissue injury and in regulating leucocytes recruitment following tissue damage, which are key drivers of acute respiratory distress syndrome (ARDS), disseminated intravascular coagulation and cardiovascular complications in COVID-19 [17,18]. According to this role, a large retrospective study identified MR-proADM as an important triage marker to stratify the risk of worse prognosis in COVID-19 patients [18,19].

Although very informative, many cytokines cannot be easily assayed in daily practice, and we do not know exactly in which relationship they are with the classic inflammatory biomarkers. The main aim of this retrospective study conducted in a large cohort of COVID-19 patients hospitalized during the first and second waves, was to identify the most relevant cytokines which, when assessed in the earliest stages following hospital admission, may help to select subjects with worse prognosis who need most intensive care and possibly suggest more targeted intervention.

## 2. Results

### 2.1. Patients Features and Correlation between Biomarkers and Clinical Outcome

We enrolled 415 consecutive COVID-19 patients (272 males and 143 females; mean age 67.7 ± 14.2 years) admitted to our hospital in the first and second waves of the pandemic for whom all the biomarkers under study were tested in the same sample within 72 h from admission. The demographic, laboratory and clinical characteristics of patients are illustrated in Table 1. 

As regards the comorbidities, patients were affected by obesity (58.3%, 242/415), hypertension (56.6%, 235/415), cardiovascular disease (35.7%, 148/415), dyslipidemia (24.1%, 100/415), diabetes (20%, 83/415), solid and hematologic neoplasms (12.3%, 51/415), chronic kidney diseases (12%, 50/415), chronic obstructive pulmonary disease (8.7%, 36/415), autoimmune diseases (6.5%, 27/415), liver disease (6%, 25/415), primary or secondary immunosuppression (6%, 25/415).

The overall mortality rate of the cohort of patients was 65/415 (15.7%); 60/415 (14.5%) of patients during hospitalization underwent orotracheal intubation (OTI) and overall, a negative outcome (which include death and/or OTI) affected 98/415 (23.6%) of cases.

Patients showed a large variability in the concentration of CRP, MR-proADM and cytokines, well representing the different phases of the pathological process and/or disease severity. As illustrated in the correlation matrix (Figure 1), the strongest correlations appeared between IL-6 and IL-10 (Spearman r = 0.54; *p* < 0.001), IL-10 and IP10 (r = 0.69; *p* < 0.001), TNFα and IL-8 (r = 0.52, *p* < 0.001) and MR-proADM and sIL2Rα (r = 0.54, *p* < 0.001). All the other analytes showed moderate degree of correlation. CCI and age were strongly correlated (r = 0.79; *p* < 0.001) and they both showed the strongest correlation with MR-proADM (r = 0.68 and 0.61 respectively, both *p* < 0.001). 

### 2.2. Cytokines and Inflammatory Markers in Patients with Different Disease Severity

Data about the WHO score to scale the severity of lung involvement were available in 372 patients, 146 displayed WHO < 3 (39.2%) and 226 WHO ≥ 3 (60.8%). As expected, patients with WHO ≥ 3 were more males (67.7% vs. 57.5%, *p* = 0.046), more aged (median 72.1 vs. 66.8 yrs, *p* = 0.006) and presented higher CCI (*p* = 0.004) than those with WHO < 3 (Table 2). Accordingly, inflammatory markers such as CRP and MR-proADM were significantly more elevated in patients with WHO ≥ 3 (*p* < 0.0001). As regards the cytokines panel, all but IFNγ and TNFα showed increased serum concentration in COVID-19 patients with more severe pneumonia. Thus, the inflammatory markers under study represent very well the severity of the pulmonary picture even in the patient population where the WHO score was not available.

### 2.3. Cytokines and Inflammatory Markers in Patients with Different Clinical Outcomes

In order to be able to profile the biomarkers in patients with negative outcome we first analyzed all the variables under study by univariate tests comparing the 3 subgroups of patients: OTI, deceased and the combination (negative outcome) compared to those who disclosed an overall more favorable outcome.

Compared to the non-OTI, the 60 OTI patients did not differ for age and CCI (Table 3), while the percentage of males was higher (83.3% vs. 62.5%, *p* = 0.002). Thus, male sex appeared as the main demographic variable associated to OTI. As regards the laboratory markers, CRP and MR-proADM were significantly more elevated in OTI patients, as well as the majority of cytokines, with exception of IFNγ, IL1β and TNFα.

COVID-19 patients who did not survive (Table 4), were older (median age 77.6 vs. 67.4 years, *p* < 0.001) and more affected by other diseases (CCI median 3 versus 5.5; *p* < 0.001) than those who survived, but they were not different for gender (males 67.7% vs. 65.1% *p* = ns). Thus, in contrast to OTI, death was strictly correlated to age and CCI, independently from sex. Similar to OTI patients, inflammatory markers such as CRP and MR-proADM were significantly more elevated in deceased patients, as well as the majority of cytokines, again with exception of IFNγ, IL1β and TNFα (Table 4).

Finally, COVID-19 patients displaying a negative outcome (death and/or OTI), were characterized by older age (*p* < 0.0001) and higher CCI (*p* < 0.0001) compared to those with a favorable outcome (Table 5), but again they were not different for gender. In addition, in this subgroup, CRP and MR-proADM were significantly more elevated, like all other cytokines, again with the exception of IFNγ, IL1β and TNFα (Table 5).

### 2.4. Ranking Test Analysis

In order to understand the degree of information harbored in each variable with respect to disease severity and clinical outcomes, we performed ranking tests. As illustrated in Figure 2A, the best variable to discriminate between patients with WHO < 3 versus ≥ 3, was IP10, followed by CRP, sIL2Rα, IL-6, age and CCI. Notably, IL1β presented higher F-values than MR-proADM, TNFα, IL-10, IFNγ and IL-8, which were instead not significant.

As regard the need for mechanical ventilation (Figure 2B), the ranking test showed the highest F-values again for CRP, followed by IP10, male sex and IL-6, while all the other variables were described as not informative.

In contrast, multiple comorbidities (CCI) appeared as the most important variable in the ranking test for death (Figure 2C). Notably, following CCI, the biomarkers which disclosed the best F-values were IP10 and sIL2Rα, which appeared behind CRP and MR-proADM. Among the other cytokines, IL-6 and IL-10 showed significant ranking values, while IL-8, IL-1β, TNFα and IFNγ were described as not informative.

Finally, when considering the comprehensive negative outcome, the most important variables in the ranking test were IP10 and IL-6, which stayed behind CRP and MR-proADM (Figure 2D). Among the other cytokines, sIL2Rα and IL-10 showed significant ranking values, while IL-8, IL-1β, TNFα and IFNγ were again not informative (Figure 2D). Regarding the demographics, CCI and age were very important, but not the gender.

### 2.5. Decision Tree Building Up

With the aim of summarizing the data obtained so far in a prognostic model to assess the risk of a negative outcome and to assess whether the best laboratory markers found in the ranking tests contained overlapping or synergic information, a classification and regression trees (CART) model was applied to the database. The focus was made exclusively on laboratory data, thus not including age, sex and CCI. We discarded the irrelevant markers resulting from ranking tests (TNFα, IL-8, IL-1β, IFNγ), thus including the followings: sIL2Rα, IL-6, IL-10, IP10, MR-proADM and CRP. With these variables we built up a tree to classify the patients as at risk for negative outcome (OTI and/or death; Figure 3). Tree modeling identify IL-6 as the starting marker with a cut-off of 114 pg/mL and then showed good synergy between IL-10 and MR-proADM, in increasing predictive value to classify a negative outcome risk (Figure 3, right arm). On the other side, a good synergy was found between IP10, sIL2Rα and CRP in increasing predictive value to classify a non-negative outcome risk (Figure 3; left arm). These analyses allowed to identify also preliminary cut-offs for the biomarker concentrations to be validated in future prospective studies. 

## 3. Discussion

Dysregulation of the immune system, with the co-existence of both pro-inflammatory and anti-inflammatory mediators dictate the outcome of COVID-19 patients. Although the increase of circulating cytokine levels that COVID-19 patients experience is commonly referred to as cytokine storm, these patients have unique features distinguishing them from those affected by ARDS secondary to other infective and non-infective conditions [20]. Moreover, it has been reported that the levels of IL-6 found in the blood of COVID-19 patients are several folds lower than those of patients experiencing a cytokine release syndrome as a consequence of other conditions (e.g., chimeric antigen receptor T cell infusion) [20]. Indeed, it has been suggested that SARS-CoV2 may alter the immune response, impairing the antiviral activity of type I IFN, triggering hyperinflammation associated with an immunosuppressive state, resulting in a condition of immunoparalysis that correlates with adverse outcome [20].

Given these complexities, the role of cytokine monitoring in the routine follow-up of patients is still a matter of debate. In this study, a large retrospective evaluation of a definite panel of pro- and anti-inflammatory markers allowed us to build risk classification tree that may help clinicians to distinguish, as early as possible from hospital admission, patients at higher risk for the worst outcome. Cytokines showed synergic role when combined to a classic biomarker of systemic inflammation (CRP) and to an emerging marker of endothelial dysfunction (MR-proADM) and may have additive effect to better explain disease pathogenesis and suggest more targeted intervention.

As repeatedly demonstrated in COVID-19 patients, we confirmed that older age, male sex and multiple comorbidities are the most important demographic and clinical features associated to a bad prognosis. Nonetheless, these differences are not always accountable for an unfavorable patient prognosis, especially in the early stages of disease [21]. In addition, this is even more evident with the new COVID-19 variants that prove to affect even younger people and potentially less affected by concomitant pathologies or in which such pathologies may be underdiagnosed [22]. What triggers an uncontrollable progression towards an hyperinflammatory state in some subjects is still not completely understood and, above all, therapeutic strategies able to mitigate the upstream inflammatory stimulus in an effective way are still lacking.

Cytokines interplay is very complicated, since they have often multiple, redundant and unpredictable roles. However, they already represent a very good target of numerous biological therapies in all the chronic inflammatory immune-mediated diseases. Some of these biological therapies, such as IL-6, IL-1 and TNF inhibitors, showed promising results also in COVID-19 patients [23,24,25]. What seems increasingly important is that the more you act early and upstream of the inflammatory burst in the single patient, the more you have the possibility of turning it off quickly and definitively, avoiding irreversible damage in the target organs.

Considering the complexity of the inflammatory and immune response to SARS-CoV2 infection, a single marker does not appear to describe sufficiently the pathologic scenario. For this reason, we performed multiplex analysis to describe the multifaceted disorder and possibly suggest targeted intervention, as quickly as possible. The statistical analysis here described was aimed to find the best marker combination able to stratify patient prognosis. Univariate tests always identified CRP as one of the most informative players in the classification of COVID-19 risk and severity and this is in accordance with a great deal of evidence from the literature [26]. Moreover, its reliability in the routine diagnostic process made it a fundamental tool

As regards the need for mechanical ventilation, that appeared strictly linked to male sex in our series, but independent from age and CCI, CRP was the best laboratory marker selected by ranking test, followed by IP10 and IL-6. As regards the death outcome, the most informative variable was CCI, which confirms the importance of multiple comorbidities in severely compromising the course of this infection. Following CCI, the best laboratory marker appeared the increased level of IP10 that represent the interferon signature of COVID-19 and can serve as a signal of an excess, not efficacious, anti-viral response, that may lead to lung fibrosis and need for mechanical ventilation [27]. Of note, IP10 was selected in first place by the ranking test either as regard the disease severity (WHO > 3) and when considering the combined negative outcome. Following IP-10, another important biomarker selected by ranking tests was the extreme increase of sIL2Rα concentration, that in the context of COVID-19, represents the later phase of the immune paralysis, when lymphocytes were incapable to do their job against the virus [6]. sIL2Rα elevation is also observed in haemophagocytic lymphohistiocytosis (HLH), a life-threatening condition characterized by fever, hyperferritinaemia, progressive cytopenia and multi-organ dysfunction coupled with macrophage activation. Although many features of HLH overlap with severe COVID-19, an accurate esteem of the frequency of SARS-CoV2 triggered HLH is still lacking [28]. 

Closely related to CRP is IL-6, the pivotal cytokine in the inflammatory process, with pleiotropic activity downstream of the stimulus [29]. IL-6 is now available on several automated diagnostic platforms and can be easily analyzed even in an emergency and the targeted reduction of its activity by Tocilizumab has been demonstrated a good opportunity in COVID-19 patients [24]. Notably, IL-6 proves to be the entry marker in the decision tree build up for negative outcome in our series, with a cut-off of 114 pg/mL, that is in line with previous reports [30]. Downstream of IL-6, IL-10 with a concentration >17.5 pg/mL and MR-proADM, with a cut-off >1.245 mmol/L, discriminate a small subset of patients (n. 23), 91% of which underwent OTI and/or death. 

The paradoxical negative impact of increased IL-10 on patient outcome has been viewed as either resistance to the anti-inflammatory effect of the cytokine, especially in the diabetic setting, or to the pro-inflammatory effect of high concentrations of IL-10 [13]. On the other side of the tree, when IL-6 is still elevated but lower than 114 pg/mL, IP10 is <1628 pg/mL and IL-2Rα is <5308, a large proportion of patients (217/317, 68%) with a better outcome was identified. 

Thus, our data suggest that, in the early phase of the phlogistic process, when IL6 may be already very high, the focus must be made on the degree of immune paralysis (sIL2Rα) and on the anti-inflammatory response (IL-10). It seems that, when sIL2Rα elevation is coupled with maximal elevation of IL-10 and IL-6 (two cytokines able to reduce NK cell cytotoxicity, which is another important feature of HLH), the risk of death is very high [28]. The presence of vascular-dependent organ damage, as represented by increased level (at least >1.200 mmol/L) of MR-proADM, is also a fundamental signal of bad prognosis. All these concepts are recapitulated in the decisonal tree.

The present paper has some limits. First of all, it is a retrospective study and needs to be confirmed in a second independent series and in prospective studies, both of which are ongoing in our hospital. Another limitation of this study is the relatively small number of patients that was included compared to the large number of COVID-19 patients that could have been evaluated. Nevertheless, the cohort under study represents an excellent cross-section of the COVID-19 population that was admitted to our hospital during the first two pandemic waves. Patients displayed the full spectrum of COVID-19 disease severity ranging from mild, self-limiting respiratory tract illness, to severe progressive pneumonia, multi-organ failure and death. Another limit may be the use of a single value of cytokines level instead of their kinetics during the whole hospitalization period. However, we chose the levels of cytokines in the early stages from hospital admission since our focus was the prognostic value towards the following outcome of patients. 

The originality of this work is represented by the combined analysis of cytokines (pro and anti-inflammatory), basic inflammatory tools (CRP) and a marker of endothelial damage (MR-proADM). This combination may give a more comprehensive view of the pathological process undergoing in the single patient, also considering endothelial damage due to cardiovascular and/or metabolic syndromes affecting the patient. In fact, MR-proADM resulted strictly correlated to age and CCI, that are key players in determining an unfavorable outcome in COVID-19 infection. 

Moreover, data obtained in the COVID-19 context may also be useful in other infections when the altered response of the patient may be detrimental. Finally, we not only identify alarming cut-offs to early identify patients at higher risk, but we can also suggest a biomarker profile that will allow a patient to be safely discharged.

Another important point of our work is represented by the feasibility of the dosages of the identified markers in daily clinical practice, which is crucial in order to translate what we have achieved to the patient’s bed. At present, all the cytokines can be assayed in our laboratory with a turn-around time of less than 48 h.

At the moment, therapeutic guidelines for COVID-19 recommend steroids for all patients with symptoms for more than 7 days and requiring oxygen therapy, irrespective the immune status. According to our results and thanks to the availability of multiple cytokines in the same way as the classic laboratory biomarkers, probably we will be able in the near future, to treat only patients who really need immune-suppressive agents, deserving different therapeutic approaches to those displaying features of immunoparalysis. In addition, this will be translated to other pathologic contexts. For instance, it was recently reported that during the early stages of Staphylococcus aureus bacteremia, increased levels of IL-10 are related to persistent bacteremia and higher mortality, while increased levels of IL-1β, after three to seven days of antibiotic therapy, are related to survival [31].

## 4. Materials and Methods

### 4.1. Patients

This retrospective observational study was conducted at the University Hospital of Udine, a tertiary acute care regional hospital serving an area of more than 700,000 people in the northeast of Italy. We enrolled consecutive patients with confirmed diagnosis of SARS-CoV2 infection by at least one positive nasopharyngeal swab. Patients were hospitalized between May 2020 and March 2021 at the Infectious Diseases Clinic of our Academic Hospital including a regular ward and a subintensive care unit. The Charlson Comorbidity Index (CCI) was collected in all patients. The clinical severity of patients was evaluated using the classification reported on World Health Organization (WHO) guidance [32]. All research was performed in accordance with the relevant guidelines and regulations. Patients were enrolled in accordance with the Helsinki Declaration and the study was approved by our Institutional Review Board (MANDI registry—Unique Protocol ID: 3929). At hospital admission, patients were routinely asked to consent to anonymized aggregate data for research purposes through the General Electronic COnsents (GECO system). The manuscript was drafted according to the Standards for the Reporting of Diagnostic accuracy studies STARD criteria [33].

### 4.2. Laboratory Analysis

SARS-CoV2 infection detection on nasopharyngeal swabs was based on the presence of unique sequences of virus RNA by nucleic acid amplification in real-time PCR (RT-PCR). Genes investigated were the E gene for screening and the RdRp and N genes for confirmation. RT-PCR was performed using a LightMix^®^ Modular SARS and Wuhan CoV E-gene kit on a LightCycler^®^ 480 II instrument (Roche, Basel, Switzerland). The specimens were considered positive if the cycle threshold (Ct) value for at least one of the three genes was ≤36. The RT-PCR was conducted as recommended by the World Health Organization for COVID-19 clinical management and outbreak control purposes [34]. Within the first 72 h from hospital admission, patients were tested for a wide panel of serum biomarkers, including C-reactive protein (CRP), Mid-regional proadrenomedullin (MR-proADM), Interferon gamma (IFNγ), interleukin 6 (IL-6), IL1-β, sIL2Rα/sCD25, IL-8, IL-10, IP10/CXCL10 and TNFα, CRP was tested by a diagnostic method (Elecsys, Roche Diagnostics, Basel, Switzerland) and MR-proADM plasma concentrations were measured in an automated Kryptor analyzer, using TRACE technology (Kryptor, BRAHMS, Hamburg, Germany). All cytokines were analyzed using a microfluidic ultrasensitive ELISA using the Protein simple plex technology on ELLA instrument (R&D systems, Biotechne, Minneapolis, MN, USA). 

### 4.3. Statistical Analyses

Baseline patient characteristics were summarized using standard descriptive statistics, with number and percentages for binary and categorical outcomes and appropriate measures for continuous outcomes (e.g., mean ± standard deviation or median and interquartile range, depending on their distribution). Data analysis was performed using Python 3 in Jupyterlab environment. Modules employed are pandas and numpy for database manipulation, seaborn for visualization (boxplot, violinplot, scatterplot, heatmap, histplot), statsmodels for statistical analysis (ttest_ind), scikit-learn for ranking (f_classif) and modeling (tree) [35]. Correlations between all the variables considered in this study were investigated using a correlation matrix with Spearman method, being more robust than Pearson against outliers. The hypothesis testing was performed using the Mann–Whitney method, since the distribution did not pass the normality test. Documentation of employed methods is available online. In Tree modeling the hyperparameters were the followings: {‘criterion’: ‘gini’, ‘max_depth’: 3, ‘max_features’: None, ‘max_leaf_nodes’: None, ‘min_impurity_decrease’: 0.0, ‘min_impurity_split’: None, ‘min_samples_leaf’: 10, ‘min_samples_split’: 10, ‘min_weight_fraction_leaf’: 0.0, ‘splitter’: ‘best’}.

## 5. Conclusions

In conclusion, besides the well-known inflammatory markers CRP and IL-6, a panel of other cytokines such as IP10, sIL2Rα and IL-10 appear as very interesting tools to stratify COVID-19 patients, deserving possible new strategic intervention. 

## Figures and Tables

**Figure 1 ijms-23-04830-f001:**
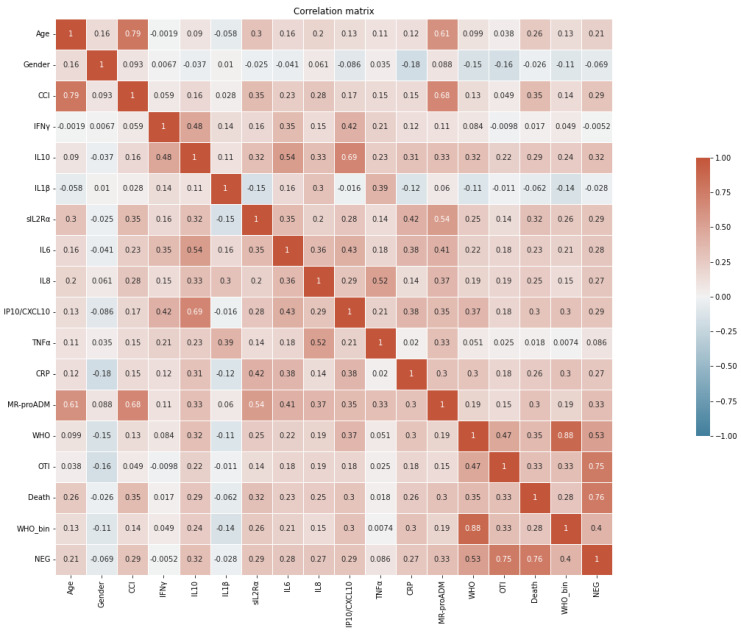
Correlation matrix. Heat map summarizing the results of Spearman correlation analysis among the demographic, clinical and cytokine data considered. Correlation coefficients (r) are shown in the relative boxes found at the intersection between the considered variables. Positive correlation coefficients are shown in shadows of red, while negative correlations in shadows of blue.

**Figure 2 ijms-23-04830-f002:**
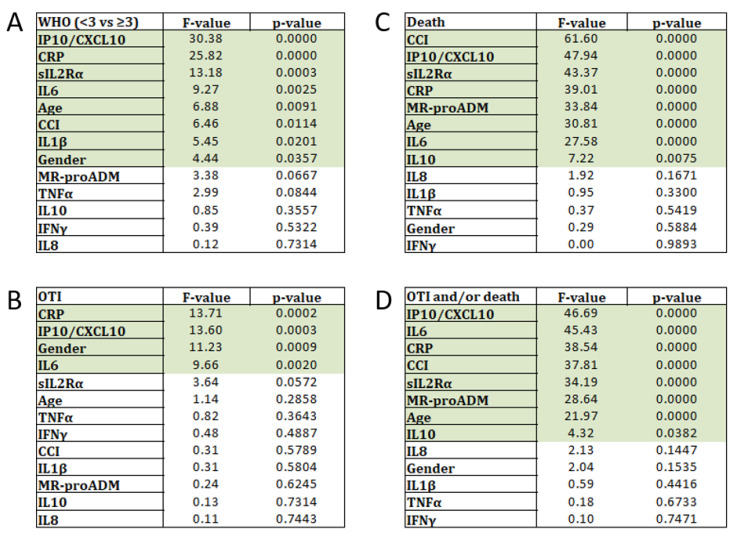
Ranking tests results. (**A**) In panel (**A**) are illustrated the results of the ranking test applied to select the best variables to discriminate between patients with WHO < 3 versus ≥ 3. (**B**) In panel (**B**), are illustrated the results of the ranking test to select the best variable to discriminate between patients who needed or not OTI; (**C**) in panel (**C**), the results to discriminate between patients who deceased or not and finally, (**D**) in panel (**D**), the best variables to discriminate between patients who have generally had a negative outcome or not. Significant markers are shown in shadows of green.

**Figure 3 ijms-23-04830-f003:**
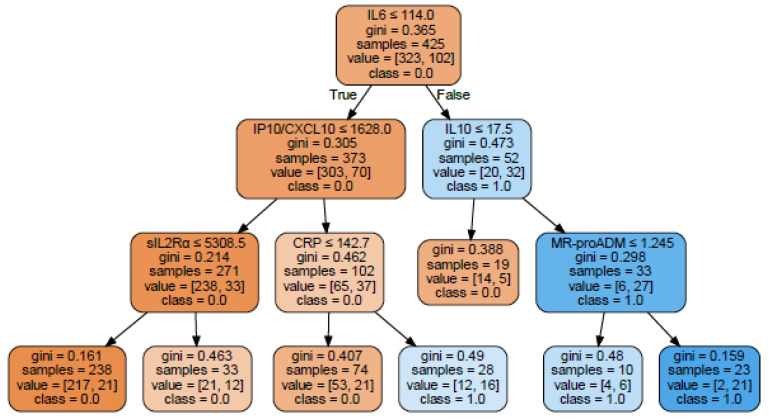
Classification tree for negative outcome risk. Classification tree with negative outcome (OTI and/or death) as target variable. Colour intensity is proportional to the purity of each node/leaf: orange for class 0.0 (not negative outcome), blue for class 1.0 (negative outcome). In each leaf, top to bottom: name of the variable with cut-off for the subsequent population split (True vs. False); Gini index of the node; number of samples contained in the leaf; number of patients in class 0, 1 (value = [number of patients in class 0, number of patients in class 1]); dominant class (e.g., Class = 0.0). Gini index is a measure of “impurity” of the node, with pure nodes having Gini index of 0. The algorithm splits nodes based on the minimum value of Gini index.

**Table 1 ijms-23-04830-t001:** Demographic, clinical and laboratory data of the study patients. SD = standard deviation; WHO score (World Health Organization; data were available in 372 patients); Charlson Comorbidity Index (CCI); CRP: C-reactive protein; MR-proADM: mid-regional proadrenomedullin (normal range < 0.56 nmol/L). IFNγ normal range < 0.99 pg/mL; IL-10 normal range: 1.8–3.8 pg/mL; IL-1β normal range: <0.16 pg/mL; sIL2Rα normal range 440–1435 pg/mL; IL-6 normal range: <7 pg/mL; IL-8 normal range: 6.7–16.2 pg/mL; IP10 normal range: 37.2–222 pg/mL; TNFα normal range 7.8–12.2 pg/mL.

Variables	Mean	SD	Median	Percentile 25	Percentile 75
Age (yrs)	67.7	14.2	69.5	57.3	78.6
WHO	2.7	1.2	3.0	2.0	4.0
CCI	3.5	2.4	3.0	2.0	5.0
IFNγ (pg/mL)	5.6	14.7	1.6	0.3	4.9
IL-10 (pg/mL)	35.5	335.1	13.8	7.7	22.5
IL-1β (pg/mL)	0.3	0.6	0.2	0.0	0.4
sIL2Rα (pg/mL)	3875.0	2392.0	3314.0	2373.0	4547.0
IL-6 (pg/mL)	69.6	137.2	31.7	15.9	63.0
IL-8 (pg/mL)	58.0	95.6	39.5	26.6	59.5
IP10 (pg/mL)	1374.0	958.0	1216.0	669.0	1838.0
TNFα (pg/mL)	27.0	21.9	26.9	16.0	29.0
CRP (mg/L)	79.6	63.5	68.6	32.9	106.4
MR-proADM (nmol/L)	1.4	1.4	1.0	0.8	1.4

**Table 2 ijms-23-04830-t002:** Demographics, clinical and laboratory markers in patients with different disease severity. WHO data were available in 372 patients. WHO score goes from 0 to 6 as follows: 0 = asymptomatic; 1 = mild symptoms; 2 = moderate pneumonia symptoms; 3 = severe symptoms of pneumonia without acute respiratory distress syndrome (ARDS); 4 = critically ill with ARDS; 5 = critically ill with sepsis; 6 = critically ill with septic shock.

Variables		WHO ≥ 3		*p* Value
		No	Yes	
Sex	M	84 (57.5%)	153 (67.7%)	0.046 *
	F	62 (42.5%)	73 (32.3%)	
Age (yrs)		66.8 (54.7–77.7)	72.1 (62.3–79.1)	0.006 °
CCI		3 (1–4)	3 (2–5)	0.004 °
IFNγ (pg/mL)		1.4 (0.3–4.9)	1.7 (0.4–4.5)	0.499 °
IL-10 (pg/mL)		9.6 (6.1–18.7)	16.4 (9–23.8)	<0.001 °
IL-1β (pg/mL)		0.2 (0–0.5)	0.1 (0–0.3)	0.002 °
sIL2Rα (pg/mL)		2861 (1956–4174)	3698.5 (2796–5002)	<0.001 °
IL-6 (pg/mL)		27.7 (9.3–55.3)	36 (22–70.3)	<0.001 °
IL-8 (pg/mL)		32.2 (21.5–51.8)	43 (28.4–60)	0.004 °
IP10 (pg/mL)		902.5 (435–1419)	1395 (909–1986)	<0.001 °
TNFα (pg/mL)		28.3 (4.8–170)	24.7 (3.9–137)	0.084
CRP (mg/L)		44.3 (13.3–90.7)	80.1 (48.1–129.8)	<0.001 °
MR-proADM (nmol/L)		0.9 (0.7–1.3)	1.1 (0.9–1.5)	<0.001 °

* Chi square test; ° Mann-Whitney test.

**Table 3 ijms-23-04830-t003:** Demographics, clinical and laboratory markers in OTI and non-OTI patients.

Variables		OTI		*p* Value
		No	Yes	
Sex	M	222 (62.5%)	50 (83.3%)	0.002 *
	F	133 (37.5%)	10 (16.7%)	
Age (yrs)		68.7 (56.5–78.9)	70.6 (65.5–76.6)	0.317 °
CCI		3 (2–5)	3 (2–5)	0.319 °
IFNγ (pg/mL)		1.6 (0.3–4.7)	1.5 (0.5–6)	0.997 °
IL-10 (pg/mL)		12.7 (7.2–21.6)	19.8 (14.4–30.3)	<0.001 °
IL-1β (pg/mL)		0.2 (0–0.4)	0.2 (0–0.4)	0.752 °
sIL2Rα (pg/mL)		3292 (2298–4462)	4020 (2937–5219.5)	0.003 °
IL-6 (pg/mL)		29.7 (13.3–57.3)	44.6 (27–116)	<0.001 °
IL-8 (pg/mL)		36.6 (25.9–57.4)	52.3 (34–75.1)	<0.001 °
IP10 (pg/mL)		1138 (647–1727)	1676 (1195–2140.5)	<0.001 °
TNFα (pg/mL)		27.4 (3.9–219)	24.7 (9.4–90)	0.364
CRP (mg/L)		63.2 (29.2–102)	98.8 (57.7–148.2)	<0.001 °
MR-proADM (nmol/L)		1 (0.8–1.4)	1.2 (0.9–1.5)	0.004 °

* Chi square test; ° Mann-Whitney test.

**Table 4 ijms-23-04830-t004:** Demographics, clinical and laboratory markers in deceased and non-deceased patients.

Variables		Death		*p* Value
		No	Yes	
Sex	M	228 (65.1%)	44 (67.7%)	0.691 *
	F	122 (34.9%)	21 (32.3%)	
Age (yrs)		67.4 (55.6–77.2)	77.6 (69.4–83.7)	<0.001 °
CCI		3 (2–4)	5.5 (4–7)	<0.001 °
IFNγ (pg/mL)		1.5 (0.3–4.5)	1.7 (0.3–5.4)	0.664
IL-10 (pg/mL)		12.6 (7.2–20.7)	21.6 (13.3–34.1)	<0.001 °
IL-1β (pg/mL)		0.2 (0–0.4)	0.1 (0–0.4)	0.242
sIL2Rα (pg/mL)		3165 (2293–4107)	5072 (4015–6584)	<0.001 °
IL-6 (pg/mL)		28.9 (13–55.3)	56 (31.4–127.1)	<0.001 °
IL-8 (pg/mL)		36 (25.6–53.6)	58.4 (38.6–76.4)	<0.001 °
IP10 (pg/mL)		1108 (638–1672)	1964 (1418–2666)	<0.001 °
TNFα (pg/mL)		27.3 (4.8–219)	25.4 (3.9–91.6)	0.541
CRP (mg/L)		62.7 (28.6–100.9)	105.2 (57.1–167)	<0.001 °
MR-proADM (nmol/L)		1 (0.8–1.3)	1.4 (1–2.4)	<0.001 °

* Chi square test; ° Mann-Whitney test.

**Table 5 ijms-23-04830-t005:** Demographics, clinical and laboratory markers in patients with a negative outcome.

Variables		NEGATIVE Outcome (Death/OTI)		*p* Value
		No	Yes	
Sex	M	203 (64.0%)	69 (70.4%)	0.246 *
	F	114 (36.0%)	29 (29.6%)	
Age (yrs)		67.1 (55.3–78.2)	74.3 (67.8–81.6)	<0.001 °
CCI		3 (1–4)	4 (3–6)	<0.001 °
IFNγ (pg/mL)		1.5 (0.3–4.5)	1.6 (0.3–5.4)	0.910 °
IL-10 (pg/mL)		11.4 (6.9–19.7)	20.9 (13.3–33)	<0.001 °
IL-1β (pg/mL)		0.2 (0–0.4)	0.2 (0–0.4)	0.534 °
sIL2Rα (pg/mL)		3150 (2238–4101)	4415.5 (3087–5866)	<0.001 °
IL-6 (pg/mL)		28.6 (12.2–51.5)	46.1 (27.8–121)	<0.001 °
IL-8 (pg/mL)		34.9 (25.4–51)	55.6 (34.4–78.8)	<0.001 °
IP10 (pg/mL)		1072 (637–1578)	1754.5 (1278–2314)	<0.001 °
TNFα (pg/mL)		27.3 (4.8–219)	26.2 (3.9–91.6)	0.673
CRP (mg/L)		60.9 (28.1–100)	99.9 (53.8–160)	<0.001 °
MR-proADM (nmol/L)		0.9 (0.7–1.3)	1.3 (1–2)	<0.001 °

* Chi square test; ° Mann-Whitney test.

## Data Availability

Clinical and laboratory data are included in the MANDI registry.

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
