# Peer review of "Cytokines from Bench to Bedside: A Retrospective Study Identifies a Definite Panel of Biomarkers to Early Assess the Risk of Negative Outcome in COVID-19 Patients"

_ijms, 2022, doi:10.3390/ijms23094830_

Round 1
Reviewer 1 Report
This study examined a panel of cytokines and other biomarkers that could be used for early assessment of patients with COVID-19. There are many reports about different prognostic biomarkers related to COVID-19 (especially for IL-6 and CRP), but the novelty of this study is that it combines different standardized biomarkers, which allows a detailed view of the undergoing pathological processes of the infected patient, including tissue damages (MR-proADM). Based on these markers, the authors suggested a classification tree for the patients at risk for a negative outcome.
The following points should be addressed:
- Abstract: Please remove the subheadings “Background, Methods, Results, Conclusions” within the abstract.
- In the current study, CRP is included as a biomarker, but there is not information about CRP in the Introduction or Discussion. There are many reports about the role of CRP in COVID-19 patients.
- In section 2.5: Please, explain the abbreviation CART model.
- Table 1. The presentation of this table is not optimal. Please, split it to two tables or give the first part (Sex, Death, OTI) as text. It will be good in the column ‘Median’ to give and ± SD. In the section 4.3. Statistical analyses, row 3 this is mentioned “(e.g. mean ± standard deviation…….)” It is not given anywhere.
- Figure 2: Please give in the title of the table what is presented in A), B), C), D).
- Figure 3: Replace “<=” with “≤”
Author Response
Response to Reviewer n. 1
We really thank the Reviewer for the suggestions that allowed us to further improve the paper. We have modified the manuscript in accordance with the requirements. I answer the questions below, point by point (in bold).
-
Abstract: Please remove the subheadings “Background, Methods, Results, Conclusions” within the abstract.
Subheadings were removed from Abstract and at line 33, "In conclusion" was added.
-
In the current study, CRP is included as a biomarker, but there is not information about CRP in the Introduction or Discussion. There are many reports about the role of CRP in COVID-19 patients.
We agree with the Reviewer, many reports underline the role of CRP in COVID-19 and we have added a reference in the Discussion (line 249). Now the references list includes one more reference and the order has changed accordingly (see reffs. 29 to 34).
-
In section 2.5: Please, explain the abbreviation CART model.
At line 151 (section 2.5) we have explain CART as classification and regression trees.
-
Table 1. The presentation of this table is not optimal. Please, split it to two tables or give the first part (Sex, Death, OTI) as text. It will be good in the column ‘Median’ to give and ± SD. In the section 4.3. Statistical analyses, row 3 this is mentioned “(e.g. mean ± standard deviation…….)” It is not given anywhere.
Thanks for the suggestion. We deleted the first part of table 1 since data about age, sex, death and OTI are already in the text. Instead, we have included also data about mean and standard deviation for all the variables.
-
Figure 2: Please give in the title of the table what is presented in A), B), C), D).
Line 187-191: we have included more detailed description of what is illustrated in each panel of figure 2.
-
Figure 3: Replace “<=” with “≤”
We replaced “<=” with “≤” in figure 3, either in the ms and in the graphical abstract.
Reviewer 2 Report
The manuscript by Fabris et al., “Cytokines from bed to bedside: a respective study identifies a definite panel of biomarkers to early assess the risk of negative outcome in COVID-19 patients” is focusing on the analysis of cytokines as biomarkers of severe COVID-19. This study is interesting because the authors suggest that the combination of biomarkers of inflammation and endothelial dysfunction represent a good biomarker to targeting an intervention. This manuscript is well written, but some points could be considered by authors to strengthen the current version.
- In title, is it correct “……a RESPECTIVE study”…?, I think “…a RESTROSPECTIVE study”… is more appropriate.
- In COVID-19 it is well know that some comorbidities like obesity are the high-risk factors to developed severe disease. Could authors include information about the more frequently comorbidities in the patients included in this study? Particularly, the obesity is considered an inflammatory disease. How impact this “basal inflammation” in the tree for negative outcome risk suggested.
- The IL-6 response and patient outcomes could be affected by medications, such as corticosteroids. What was the treatment the patients received during this study?
Author Response
Response to Reviewer n. 2
We really thank the Reviewer for the suggestions that allowed us to further improve the paper. We have modified the manuscript in accordance with the requirements. I answer the questions below, point by point (in bold).
-
In title, is it correct “……a RESPECTIVE study”…?, I think “…a RESTROSPECTIVE study”… is more appropriate.
Sorry for the mistake, yes is RETROSPECTIVE, we corrected the title. But also corrected another very funny mistake: from bench (not from bed) to bedside! Very sorry for these mistakes.
-
In COVID-19 it is well know that some comorbidities like obesity are the high-risk factors to developed severe disease. Could authors include information about the more frequently comorbidities in the patients included in this study? Particularly, the obesity is considered an inflammatory disease. How impact this “basal inflammation” in the tree for negative outcome risk suggested.
As regards the comorbidities, patients were affected by obesity (58.3%, 242/415), hypertension (56.6%, 235/415), cardiovascular disease (35.7%, 148/415), dyslipidemia (24.1%, 100/415), diabetes (20%, 83/415), chronic kidney diseases (12%, 50/415), chronic obstructive pulmonary disease (8.7%, 36/415), liver disease (6%, 25/415), solid and hematologic neoplasms (12.3%, 51/415), autoimmune diseases (6.5%, 27/415), primary or secondary immunosuppression (6%, 25/415). We included this description in the paper (see lines 92 - 96).
So obesity was the most frequent comorbidity, but many patients presented also hypertension, cardiovascular diseases and dyslipidemia. The correlation matrix pointed out a strong correlation between CCI and MR-proADM, so the endothelial dysfunction may represent very well the combination of such multiple comorbidities. It's difficult to assess exactly how the "basal inflammation" related to obesity may impact on the tree. From the literature we know that obesity is associated with over production of acute-phase reactants, such as IL-6, IL-8, TNF-α, C-reactive protein (CRP), and monocyte chemotactic protein-1. The significant increase of inflammatory biomarkers, such as IL-6 and CRP in severe COVID-19 patients, are directly or indirectly linked to adipocytes with sub-clinical low-grade inflammation that further exacerbates COVID-19 severity in individuals with obesity. Adipocytes secrete leptin that could play a significant role in developing severe COVID-19 infection in patients with obesity. IL-6 stays at the top of our tree. So we think that IL-6 may represent the best marker related also to obesity. But, not all obese patients develop severe COVID-19. And the same was true for aged people. For this reason we decided to put on the tree only the biomarkers, to better understand how they can profile the dysregulated inflammatory response triggered by SARS-CoV2 in some patients, independently from age and CCI. Our focus was to assess if and which cytokines may be of real utility beside the well known employed biomarkers to better profile a patient and to suggest different therapeutic approaches (i.e. not immunosuppressive if the patient displays the so-called immunoparalysis).
-
The IL-6 response and patient outcomes could be affected by medications, such as corticosteroids. What was the treatment the patients received during this study?
Cytokines were tested as early as possible after hospital admission. Then patients were treated according to the standard of care in that moment as follows: dexamethasone in 68.7% (284/415) of cases and antiviral therapy (remdesivir) in 16.7%(70/415) of cases. Only 2 patients (0.5%) were treated with tocilizumab.